# Breast Cancer Patient’s Outcomes after Neoadjuvant Chemotherapy and Surgery at 5 and 10 Years for Stage II–III Disease

**DOI:** 10.3390/cancers16132421

**Published:** 2024-06-30

**Authors:** Catalina Falo, Juan Azcarate, Sergi Fernandez-Gonzalez, Xavier Perez, Ana Petit, Héctor Perez, Andrea Vethencourt, Silvia Vazquez, Maria Laplana, Miriam Ales, Agostina Stradella, Bartomeu Fullana, Maria J. Pla, Anna Gumà, Raul Ortega, Mar Varela, Diana Pérez, Jose Luis Ponton, Sara Cobo, Ana Benitez, Miriam Campos, Adela Fernández, Rafael Villanueva, Veronica Obadia, Sabela Recalde, Teresa Soler-Monsó, Ana Lopez-Ojeda, Evelyn Martinez, Jordi Ponce, Sonia Pernas, Miguel Gil-Gil, Amparo Garcia-Tejedor

**Affiliations:** 1Multidisciplinary Breast Cancer Unit, Department of Medical Oncology, Institut Català d’Oncologia, 08908 Barcelona, Spain; acvethencourt@iconcologia.net (A.V.); silviavazquez@iconcologia.net (S.V.); astradella@iconcologia.net (A.S.); tomeu.fg9@gmail.com (B.F.); afernandezortega@iconcologia.net (A.F.); ravillanueva@iconcologia.net (R.V.); veronica.obadia@iconcologia.net (V.O.); srecalde@iconcologia.net (S.R.); spernas@iconcologia.net (S.P.); mgilgil@iconcologia.net (M.G.-G.); 2Instituto de Investigación Biomédica de Bellvitge (IDIBELL), 08908 Barcelona, Spain; sfernandezg@bellvitgehospital.cat (S.F.-G.); apetit@bellvitgehospital.cat (A.P.); hpmontero@iconcologia.net (H.P.); maria.laplana.torres@gmail.com (M.L.); mjpla@bellvitgehospital.cat (M.J.P.); anna.guma@bellvitgehospital.cat (A.G.); rortega@bellvitgehospital.cat (R.O.); mvarela@iconcologia.net (M.V.); emperez@iconcologia.net (E.M.); jponce@bellvitgehospital.cat (J.P.); agarciat@bellvitgehospital.cat (A.G.-T.); 3Multidisciplinary Breast Cancer Unit, Department of Pathology, Hospital Universitari Bellvitge, 08907 Barcelona, Spain; jazcarate@bellvitgehospital.cat (J.A.); tsoler@bellvitgehospital.cat (T.S.-M.); 4Multidisciplinary Breast Cancer Unit, Department of Gynecology, Hospital Universitari Bellvitge, 08907 Barcelona, Spain; mcampos@bellvitgehospital.cat; 5Information and Data Analysis Department, Institut Català d’Oncologia, Bellvitge Research Institute, 08908 Barcelona, Spain; fjperez@iconcologia.net (X.P.); jponton@iconcologia.net (J.L.P.); 6Multidisciplinary Breast Cancer Unit, Department of Radiotherapy, Institut Català d’Oncologia, 08908 Barcelona, Spain; 7Multidisciplinary Breast Cancer Unit, Department of Radiology, Hospital Universitari Bellvitge, 08907 Barcelona, Spain; 8Multidisciplinary Breast Cancer Unit, Department of Reparative Surgery, Hospital Universitari Bellvitge, 08907 Barcelona, Spain; dperez@bellvitgehospital.cat (D.P.); ablopez@bellvitgehospital.cat (A.L.-O.); 9Multidisciplinary Breast Cancer Unit, Department of Pharmacy, Hospital Universitari Bellvitge, 08907 Barcelona, Spain; scobo@bellvitgehospital.cat; 10Multidisciplinary Breast Cancer Unit, Department of Nuclear Medicine, Hospital Universitari Bellvitge, 08907 Barcelona, Spain; abenitez@bellvitgehospital.cat

**Keywords:** neoadjuvant chemotherapy, breast cancer, survival, prognostic factors, well-informed decision making

## Abstract

**Simple Summary:**

Neoadjuvant chemotherapy in breast cancer facilitates breast and axillary surgery and offers significant prognostic value. We present a retrospective cohort of 482 stage II and III breast cancer patients treated with neoadjuvant chemotherapy based on anthracycline and taxans, plus antiHER2 in HER2-positive cases. The 10-year estimated disease free survival was 77.3% (95%CI 73.3–81.4%) and the Breast cancer specific survival 83.7% (95%CI 80.3–87.2%). The statistically independent factors related to patient survival were pathology subtype (lobular cancers HR, 4); molecular surrogate subtype (triple negative HR, 4); type of surgery (mastectomy HR, 2), response to chemotherapy (the risk incremented according to the residual cancer burden in 2.2, 4.4 and 8.0 times in I, II and III, respectively) and vascular invasion (HR, 2.4). BRCA carriers presented a longer survival, with an estimated 10 years DDFS of 89.6% vs. 77.2% for non-carriers, *p* = 0.054. Long-term outcomes after neoadjuvant chemotherapy can help patients and clinicians make well-informed decisions.

**Abstract:**

**Introduction**: Neoadjuvant chemotherapy in breast cancer offers the possibility to facilitate breast and axillary surgery; it is a test of chemosensibility in vivo with significant prognostic value and may be used to tailor adjuvant treatment according to the response. **Material and Methods**: A retrospective single-institution cohort of 482 stage II and III breast cancer patients treated with neoadjuvant chemotherapy based on anthracycline and taxans, plus antiHEr2 in Her2-positive cases, was studied. Survival was calculated at 5 and 10 years. Kaplan–Meier curves with a log-rank test were calculated for differences according to age, BRCA status, menopausal status, TNM, pathological and molecular surrogate subtype, 20% TIL cut-off, surgical procedure, response to chemotherapy and the presence of vascular invasion. **Results**: The pCR rate was 25.3% and was greater in HER2 (51.3%) and TNBC (31.7%) and in BRCA carriers (41.9%). The factors independently related to patient survival were pathology and molecular surrogate subtype, type of surgery, response to NACT and vascular invasion. BRCA status was a protective prognostic factor without reaching statistical significance, with an HR 0.5 (95%CI 0.1–1.4). Mastectomy presented a double risk of distant recurrence compared to breast-conservative surgery (BCS), supporting BCS as a safe option after NACT. After a mean follow-up of 126 (SD 43) months, luminal tumors presented a substantial difference in survival rates calculated at 5 or 10 years (81.2% compared to 74.7%), whereas that for TNBC was 75.3 and 73.5, respectively. The greatest difference was seen according to the response in patients with pCR, who exhibited a 10 years DDFS of 95.5% vs. 72.4% for those patients without pCR, *p* < 0001. This difference was especially meaningful in TNBC: the 10 years DDFS according to an RCB of 0 to 3 was 100%, 80.6%, 69% and 49.2%, respectively, *p* < 0001. Patients with a particularly poor prognosis were those with lobular carcinomas, with a 10 years DDFS of 42.9% vs. 79.7% for ductal carcinomas, *p* = 0.001, and patients with vascular invasion at the surgical specimen, with a 10 years DDFS of 59.2% vs. 83.6% for those patients without vascular invasion, *p* < 0.001. Remarkably, BRCA carriers presented a longer survival, with an estimated 10 years DDFS of 89.6% vs. 77.2% for non-carriers, *p* = 0.054. **Conclusions**: Long-term outcomes after neoadjuvant chemotherapy can help patients and clinicians make well-informed decisions.

## 1. Introduction

Neoadjuvant or primary chemotherapy (NACT) was first developed for locally advanced, inoperable breast cancer patients to reduce tumoral volume in order to facilitate operability [1,2]. Once the classical studies of Bonnadona and Fisher established the great advantages of primary chemotherapy not only to accomplish surgical advantages but also as a prognostic tool, the use of NACT has been widely incorporated as the primary therapeutic approach in stage II and III breast cancer patients [3]. Notably, this approach has greatly facilitated the shift from mastectomy to breast-conservative surgery (BCS) as a surgical option, particularly in patients with initially diagnosed cT3-4 breast cancer. A recently publication by Tinterri et al. [4] specifically looked at patients with cT3-4 breast cancer. The aims of that study were to identify predictors of breast conservation in cT3-4 breast cancer and compare the long-term oncological outcomes between BCS and mastectomy. The authors identified the absence of vascular invasion, smaller tumor size post-NACT and the complete pathological response of the primary tumor as the key predictors for breast conservation. In addition, after a follow-up of 70 months (range, 52–185), they demonstrated that BCS post-NACT does not negatively impact long-term oncological outcomes, supporting its use as a safe option for patients with cT3-4 breast cancer. Data from the Dutch Breast Audit published by Spronk et al. [5] confirm the safety of BCS after NACT. The authors analyzed trends in the use of NACT and its impact on surgical outcomes. In this audit between 2011 and 2016, the use of NACT increased from 9% to 18%, and BCS after NACT increased from 43% to 57%. Prognostic factors associated with the invasive margin rate were lobular invasive breast cancer and a hormone-receptor-positive status. When compared with the results of upfront BCS, the authors confirmed that BCS after NACT compared to primary BCS leads to equal surgical outcomes for cT2 and improved surgical outcomes for cT3 breast cancer patients. Many other studies have consistently demonstrated that BCS does not compromise recurrence and survival rates in patients with breast cancer treated with NACT [6,7,8,9].

In this sense, the American Society of Clinical Oncology (ASCO) recommendations for neoadjuvant therapy published in 2021 [10] state that NACT offers a range of potential advantages, including downstaging of the primary tumor to bring it to operability; it can also be used to reduce the extent of local therapy in the breast and axilla, reduce delays in initiating therapy, allow for more prompt treatment of subclinical distant micrometastases and enhance the ability to evaluate in vivo the response of the tumor to particular systemic agents. In addition, the ASCO recommendations incorporate a new argument to recommend the use of neoadjuvant chemotherapy, i.e., those patients for whom residual disease may require a change in therapy in view of the results of trials that have focused on using a lack of response to neoadjuvant therapy to identify patients who have a worse prognosis and could therefore benefit from additional adjuvant treatment, as demonstrated in the CreateX trial [11] for luminal and triple-negative breast cancers or the Katherine trial [12] for HER2-positive ones. This population serves as an ideal group in whom new therapies or treatment escalation strategies should be studied. Numerous trials will inform a more personalized approach to both escalation and de-escalation using neoadjuvant therapy response pathological and genomic risk markers; patient age, health and personal preferences; the efficacy of systemic and local treatments; and, in some instances, tumor response to preoperative therapy.

NACT provides the unique opportunity to assess response to treatment after months rather than years of follow-up. Achieving a pathologic complete response (pCR) has been associated with a significant improvement in disease-free survival (DFS) and overall survival (OS) and has therefore become a surrogate end point for long-term survival and a primary aim in numerous clinical trials [13,14,15,16]. A meta-analysis by Broglio et al. [17] including 36 randomized clinical trials (RTCs), including stage I-III HER2-positive breast cancer patients treated with NACT, showed a substantial improvement in event-free survival for pCR vs. non-pCR, with an HR of 0.37 (95% PI 0.32–0.43); this association was greater for patients with hormone-receptor-negative disease with an HR of 0.29 (95% PI 0.24–0.36) than for hormone-receptor-positive disease with an HR of 0.52 (95% PI 0.40–0.66). However, in a systematic review and meta-analysis, Conforti et al. [18] found a weak association between the relative risk for pCR and the hazard ratio (HR) for both disease-free survival and overall survival. In addition, most patients do not experience a complete pathological response to primary chemotherapy, the significance of lesser degrees of histological response is uncertain and the prognostic significance is unknown. Several efforts have been made to evaluate new histological grading systems, such as the Miller/Payne grading system from the University of Aberdeen [19] or the neoadjuvant response index from the Nederland [20]. Gentile et al., from Milan [21], reported the pathological response and residual tumor cellularity after neoadjuvant chemotherapy in relation to patient prognosis. These authors found statistically significant longer DFS, DDFS and OS in patients with pCR and with a residual tumor cellularity less than 40%. In 2015, the Breast International Group—North American Breast Cancer Group (the BIG-NABCG) published recommendations for the standardized pathological characterization of residual disease for neoadjuvant clinical trials [22]. Recommendations included multidisciplinary communication; clinical marking of the tumor site with clips; and radiologic, photographic or pictorial imaging of the sliced specimen to map the tissue sections and reconcile macroscopic and microscopic findings. The information required to define pCR (including carcinoma in situ or not) and residual disease stage using the current AJCC/UICC system and the Residual Cancer Burden System were recommended for the quantification of residual disease in clinical trials.

In addition, the St Gallen Consensus Conference 2023 [23] pointed out the need to offer guidance to clinicians regarding appropriate treatments for early stage breast cancer and assist in balancing the realistic trade-offs between treatment benefit and toxicity, enabling patients and clinicians to make well-informed choices through a shared decision-making process. For those reasons, the approach to breast cancer is increasingly personalized, considering specific factors such as clinical stage and biological features of the tumor, including tumor subtype and within subtype.

The present work aims to provide long-term prognostic information based on the response to neoadjuvant chemotherapy to help clinicians and patients make well-informed decisions.

## 2. Materials and Methods

A retrospective observational single-institution study was performed involving 482 stage II and III breast cancer patients who had attended the Institut Catala d’Oncologia at Bellvitge University Hospital, Barcelona, Spain, from June 2008 to December 2016.

Patients were eligible for enrollment if they were at least 18 years of age; presented with newly diagnosed, previously untreated stage II or III cancer, as determined by radiological assessment, clinical assessment or both; presented an Eastern Cooperative Oncology Group performance status score of 0 or 1 (on a 5-point scale); and showed adequate organ function. Patients were classified into five molecular surrogate subtypes according to St Gallen 2013: luminal A-like (estrogen receptor (ER) positive, progesterone receptor (PR) positive, HER2 negative and ki 67 < 20%); luminal B-like (RE positive, PR positive or negative, HER2 negative and ki 67 over 20%); luminal B Her2 (RE positive, PR positive or negative, HER2 positive); HER2 (RE negative, PR negative, HER2 positive); and triple negative (TNBC) (RE negative, PR negative and HER2). ER and PR were considered positive if expressed in 10% or more of the tumor cells. HER2-positive tumors were considered those with immunohistochemistry scores of 3+ or 2+ with gene amplification by fluorescent in situ hybridization (FISH). Patients who were already enrolled in a clinical trial were excluded from this study. Informed consent was obtained from all the patients.

NACT consisted of a 6-month anthracycline–taxane regimen plus trastuzumab in the HER2-positive cases. The anthracycline schema was doxorubicine 60 mg/m^2^ plus cyclophosphamide 600 mg/m^2^ every 21 days × 4 cycles. The taxane chosen was mainly docetaxel at 100 mg/m^2^ every 21 days × 4 cycles, but some patients received weekly paclitaxel × 12, especially for HER2 cases in combination with trastuzumab. Clinical and radiological responses were measured according to the criteria of the World Health Organization. The tumors and positive lymph nodes confirmed cytologically were marked before starting NACT with a metallic clip under ultrasound guidance to enable identification at the time of surgery. 

Patients underwent definitive surgery 3 to 4 weeks after the last cycle of NACT. Breast-conserving surgery was offered if margins were guaranteed with an optimal aesthetic result. When a mastectomy was mandatory in multicentric tumors, inflammatory cases (T4d) or if the tumor volume and breast size precluded breast conservation with satisfactory aesthetic outcomes or was offered to BRCA carriers, immediate reconstructive options were offered to the patient, predominantly with autologous flaps to avoid the development of capsular fibrosis with silicone implants. In clinical N0 cases (cN0), a sentinel lymph node biopsy was offered before chemotherapy up to 2009 and after chemotherapy thereafter, which was validated in our Breast Cancer Unit [24]. Lymphadenectomy was performed in all node-positive (N+) patients upon diagnosis and in positive sentinel lymph node cases. Target axillary dissection was introduced for cN1 patients in our institution after 2016.

pCR was defined as the absence of an infiltrating carcinoma in the breast and in the axillary lymph nodes, defined as ypT0/ypTis ypN0. For non-pCR, RCB was calculated according to the method of Symmans et al. [25]. 

Adjuvant therapy for patients with HER2-positive tumors consisted of trastuzumab for up to one year. In patients with hormone-receptor-positive tumors, adjuvant therapy consisted of hormonotherapy with tamoxifen/aromatase inhibitors for five years, extending up to 10 years in those with a high risk of recurrence such as ypT3-4 or ypN+ cases. TNBC cases with residual disease after NACT were offered adjuvant capecitabine after 2015.

Radiotherapy (RT) was administered according to our institutional protocol. RT was performed after breast-conserving surgery. A boost to the tumor bed was administered with brachytherapy or external beam RT in young patients or in those with a high risk of locoregional recurrence. RT to the chest wall after mastectomy was administered in N+ cases, those with affected surgical margins or in patients with large tumors (≥T3). Nodal RT was performed if more than 3 nodes were affected. If 1–3 nodes were involved, risk factors were taken into account to decide nodal irradiation [26]. Residual nodal disease after chemotherapy was also an indication for nodal RT. Nodal irradiation included infra and supraclavicular lymph nodes and the axilla if extended axillary fat was involved. Internal mammary lymph nodes were irradiated if affected or in N+ patients with T4 tumors or tumors located in the internal quadrants [27]. 

Follow-up was performed every 6 months from the last course of RT. Mammography was performed once a year, starting 6 months after irradiation therapy or one year from diagnosis. Other complementary examinations were performed according to the symptoms of the patient. Follow-up was extended up to 10 years in our institution, and a post-discharge program was implemented to guarantee annual control by local gynecologists that maintain a connection with or breast cancer unit. 

The following patient and tumor characteristics were analyzed: age, menopausal status, genetic test for germinal variants in genes related to hereditary cancer such as BRCA 1 or 2, body mass index (BMI), anatomic and prognostic TNM stage, pathologic subtype (ductal, lobular or others), histologic grade, hormone receptor status (positive ≥ 10% versus negative), HER2 status (positive if the ICH score is 3+ or amplified by FISH according to the 2007 and 2013 criteria of ASCO/CAP, Ki-67 (≤30 versus >30)), breast cancer molecular surrogate subtype according to St Gallen 2013 (luminal A-like, luminal B-like, luminalBHER2 positive, HER2 positive and TNBC) and the presence of tumor-infiltrating lymphocytes (TILs) at diagnosis, considered clinically meaningful at a cut-off 20% [28]. The other variables evaluated after NACT were pCR, RCB and vascular invasion. Other variables included in the data base were the type of breast surgery, type of radiotherapy, local and distant recurrences and death from any cause. 

The survival end points included disease-free survival (DFS) and distant disease-free survival (DDFS), calculated from the time of NACT commencement to a distant recurrence or death, whichever occurred first, as defined by DATECAN [29]. Overall survival (OS) was calculated from the time of NACT commencement until death from any cause, while breast-cancer-specific survival (BCSS) was calculated until death from breast cancer. The Kaplan–Meier method was used to estimate the probability of survival, while the log-rank test was applied to compare the groups. The data cut-off was 7 June 2024. Data from patients who did not have a documented event were censored at the date the patient was last known to be alive and event-free.

Categorical variables are presented as the number of cases and percentages. Continuous variables following a normal distribution are presented as means and standard deviations (SDs). Cox proportional hazards models were used to calculate hazard ratios (HR) and the 95%CI of each prognostic factor by univariate and multivariate analyses in relation to distant disease survival. A *p*-value below 0.05 was considered to indicate statistical significance. Statistical analysis was carried out using IBM SPSS version 23 (SPSS, Chicago, IL, USA).

## 3. Results 

### 3.1. Patient and Tumor Characteristics

Patient and tumor characteristics are described in Table 1. Characteristics are in accordance with a series of neoadjuvant chemotherapy factors, i.e., young age, nearly 9% BRCA carriers, one-third locally advanced tumors, mainly ductal, high proliferation rate and a limited representation of luminal A-like tumors. TILs over 20% were observed in nearly 40% of the cases. 

### 3.2. Neoadjuvant Chemotherapy Outcomes

A pathologic complete response (pCR) was achieved in 25.3% of patients. When categorized by molecular surrogate subtype, pCR was observed in 1 out of 46 (2.2%) luminal A-like, 15 out of 144 (10.4%) luminal B-like, 27 out of 91 (29.7%) luminal B Her2-positive, 40 out of 78 (51.3%) Her2-positive and in 39 out of 123 (31.7%) TNBC cases. In BRCA carriers, pCR was achieved in 18 out of 43 cases (41.9%) compared to 23.7% in non-BRCA carriers, *p* = 0.016. In those cases without pCR, the RCB was calculated with a 45% RCB III. Vascular invasion was noted in 21% of the cases. See Table 2.

Breast-conservative surgery was performed in 66% of the patients. In BRCA carriers, BCS was only performed in 51.2% of patients compared to 67.4% in non-BRCA carriers, *p* = 0.042.

### 3.3. Survival Outcomes

After a mean follow-up of 126 months (SD 43.6), 124 events were recorded: 6 contralateral cancer, 54 locoregional recurrences and 103 systemic recurrences of those 37 locoregional and systemic cases. See Table 2. 

In Figure 1, the Kaplan–Meier curves are depicted for DFS, DDFS, BCSS and OS. The 10 years estimated DFS for the whole series was 77.3% (95%CI 73.3–81.4%); that for DDFS was 76.3% (95%CI 72.3–80.3%), that for BCSS was 83.7% (95%CI 80.3–87.2%) and that for OS was 78.7% (95%CI 74.9–82.5%).

Kaplan–Meier curves and log-rank scores were calculated for DDFS according to basal characteristics, mainly age cut-off of 40, BRCA status, TNM anatomic and prognostic stage, pathology and molecular surrogate subtype and the presence of clinically meaningful TILs at a 20% cut-off. Remarkably, the 5 years DDFS differed from the 10 years DDFS, especially in the luminal subtypes: in luminal B-like cases, the 5 years DDFS was 81.2%, whereas the 10 years DDFS dropped to 74.7%, in comparison with the TNBC cases that presented a 5 years DDFS of 75.3% similar to the 10 years DDFS of 73.3%. See Figure 2.

Kaplan–Meier curves and log-rank scores were calculated according to pathological findings at surgery, mainly pCR, RCB and vascular invasion. See Figure 3.

In addition, in Figure 4, Kaplan–Meier curves are depicted according to RCB in each molecular subtype. 

### 3.4. Prognostic Factors for Patient Survival

In the univariate Cox regression model, the variables related to distant disease-free survival (DDFS) were TNM (both anatomic and prognostic), pathology subtype, the presence of TILs at a cut-off of 20% (positive related), breast-conservative surgery (positive related), the achievement of good response measured either by pCR or by RCB (positive related) and the presence of vascular invasion (inversely related). 

In the multivariate Cox regression model, the variables independently related to DDFS were the pathologic subtype, where the lobular subtype presented a four-fold higher risk of distant relapse than ductal tumors; the molecular surrogate subtype, where TNBC showed a four times higher risk for distant relapse; the decision to perform a mastectomy, which stayed independently related with a double risk of DDFS; the response to NACT by RCB, with an HR for RCBI, RCB II and RCB III in reference to RCB 0 (pCR) of 2.2, 4.4 and 8.0, respectively; and vascular invasion, with an HR of 2.4 in reference to those patients with no vascular invasion at the surgical specimen. Notably, the BRCA status nearly reached statistical significance as a protective factor, HR: 0.3 (95%CI 0.1–1.05), *p* = 0.063, in the univariate analysis and in the multivariate analysis with an HR of 0.509 (95%CI 0.18–1.43), *p* = 0.201. See Table 3. 

### 3.5. Discussion

Improvements in neoadjuvant chemotherapy regimens have led to substantial improvements in patient survival outcomes in the last 20 years [30]. The data shown in the present study from a series of stage II and III breast cancer patients treated with an anthracycline and taxan backbone schema plus antiHER in patients with Her2-positive tumors shows a clinically meaningfully difference compared to that published previously when CMF was the standard with no distinction according to HER2 status [31]. The 8 years DFS at that time was 57.63% compared to the 10 years DFS of 77.3% presented in this paper. Better odds are awaited with new pharmacological approaches such as the double antiHER2 blockade [32], antibody drug conjugates [12], the introduction of platins [33] and immune checkpoint inhibitors for TNBC [34] and the use of CDK 4 and 6 inhibitors in luminal tumors [35,36].

According to the results of the literature [13,14,15,16,17], in our series, those patients who achieved pCR showed better survival outcomes related to those patients who did not. The estimated 10 years DDFS for patients achieving pCR was 95.5% (95%CI 91.6–99.4%), and that for non-pCR was 72.4% (95%CI 67.6–77.2%), similar to results from the series of Gentile et al. [21]. 

The modern schemas selected for neoadjuvant chemotherapy in breast cancer have presented a real revolution in the last 20 years, especially in Her2-positive cancer and TNBC. The introduction of trastuzumab in the neoadjuvant setting for Her2-positive patients raised pCR from 25% up to 66.7% [37]. The double blockade with trastuzumab–pertuzumab plus chemotherapy showed a better response in survival rates compared to single agent plus chemotherapy (5 years DFS 86% for trastuzumab plus pertuzumab + docetaxel vs. 81% for trastuzumab plus docetaxel and 73% pertuzumab plus docetaxel) and has become the standard [38]. In addition, the prescription of trastuzumab emtansine (TDM-1) for those patients not achieving pCR has increased survival to 89.7% compared to 83% in the adjuvant trastuzumab group with an HR 0.60 (95%CI 0.45–0.79) [12]. In our series, Her2-positive patients presented a 5 years DDFS of 85.4% for those with RRHH-positive cancer and 87% for those with RRHH-negative cancer, similar to the Neosphere study rates. Importantly, survival rates in Her2-positive patients depended extremely on response to NACT, with a 5 years DDFS of 95% for those patients who achieved pCR compared to 50% in those patients with RCB III. Fortunately, Her2-positive patients with RCB III only represented 15% of the cases. However, these were HER2 cases that deserve more research. 

For TNBC, the therapeutical arsenal has also improved pCR and survival thanks to the introduction of platins [39,40] and immune checkpoint inhibitors [34]. The addition of platins improved pCR up to 50% [41], and the addition of pembrolizumab has raised pCR up to 65%, nearly 70%, in PDL1 positive tumors [34]. Survival has increased in accordance with the addition of platins, as was shown in the BrighTNess study that reported a 4 years DFS of 79.3% in the carboplatin arm compared to 68.5% without carboplatin, HR 0.57 (0.36–0.91) [42]. Survival data from the Keynote 522 dataset also showed a survival advantage for the arm with pembrolizumab, with an HR 0.63 (95%CI 0.43–0.93). In our study, the pCR rate in TNBC was low compared to the new standard of care for TNBC, with a 31.7% rate comparable to the schemas without platins or immune checkpoint inhibitors. However, the message of the present paper about the relevant prognostic information regarding the response to NACT is shown equally well with our data. In our series, the 5 years DDFS for TNBC was 74.1% (95%CI 66.3–82), with great differences according to the different degree of response, with those patients achieving pCR presenting an excellent survival at 5 and 10 years of 100%, whereas in RCB 1, the 5 years DDFS dropped to 80.6%, in RCB II to 69% and in RCB III to lower than 50%, log rank 0.000. These data show the imminent need to improve pCR in TNBC with new approaches such as the addition of platins, the use of immunotherapy and the approval of PARP inhibitors for BRCA carriers [43]. 

In luminal patients, NACT is not probably the best option; however, trials comparing chemotherapy with new approaches such as the combination of hormonal therapy with CDK 4/6 inhibitors have not obtained better pCR rates and were not designed to find differences in survival [44,45,46]. A different landscape has been recently opened in the adjuvant setting, where adjuvant abemaciclib for two years [35] and adjuvant ribociclib for three years [36] have demonstrated survival advantages in luminal patients with high and intermediate risk of relapse, respectively. In our series, pCR in luminal patients was as low as 2.2% in luminal A-like tumors and 10.4% in luminal B-like tumors. Interestingly, survival for luminal tumors changed significantly when calculated at 5 years or at 10 years. In luminal A tumors, the 5 years DDFs was 84.6% and dropped to 79.6% at 10 years, and in luminal B tumors, the 5 years DDFS was 81.2% and dropped to 74.7% at 10 years, with the result that distant recurrences can occur at 5 years or later whereas, in Her2 and TNBC, recurrences usually occur within the first 5 years. On the other hand, patients with luminal tumors depend less on tumor response, as already shown by P. Cortazar [47]. In our series, luminal A tumors performed well at RCB 0–2, with a value of 96.7% for the only patient that achieved pCR out of 46 cases, and for RCB III, the 5 years DDFS was 69.5%, and the 10 years DDFS dropped to 62.5%. In the luminal B-like tumors, similar results were observed, with good survival rates for RCB 0–2, whereas in those patients with RCB III, the 5 years DDFS was 72% and dropped to 65% at 10 y. A special mention is made to the scarce number of lobular carcinomas included in our series that performed extremely poorly in our study, with a 10 years DDFs of 42.9% (95%CI 16.9–68.8). A different therapeutical approach should be discovered for this special subtype of breast cancer besides chemotherapy [48]. 

The factors independently related to patient survival taken as an end point in the DDFS analysis as a surrogate of fatal outcome and with more events than breast cancer deaths were the pathologic subtype, the molecular surrogate subtype, the response to chemotherapy and vascular invasion. BRCA carriers presented a favorable outcome in relation to non-BRCA carriers, with an HR 0.5 (95%CI 0.1–1.4), but the difference was not statistically significant, *p* = 0.201, probably due to the small number of BRCA-positive cases. Lobular carcinomas presented an HR of 4 related to ductal carcinomas; TNBC an HR of 4 in reference to luminal A tumors, probably due to those tumors without response. According to RCB, HRs were 2.2, 4.4 and 8.0 for RCB I, II and III in comparison to RCB 0, and finally, those patients that presented vascular invasion in the surgical specimen presented an HR 2.33 in comparison to those patients without vascular invasion. The 10 years DDFS for patients without vascular invasion was 83.6% vs. 59.2% for patients with vascular invasion. The clinical and statistical significance of vascular invasion for patient survival deserves a special place in prognostic scores that could improve the current neoadjuvant prognostic index. 

Breast-conserving surgery was performed in 66% of this series, which included 32% of locally advanced breast cancer cases. Survival was not affected by this decision; in fact, performing a mastectomy doubled the risk of distant recurrence in comparison to BCS, *p* = 0.000. Results that confirm these findings include the already-mentioned studies from Tinterri et al. [4], the Dutch Breast Audit [5] and many others [6,7,8,9], including two recent meta-analyses [49,50]. One study including 34,169 patients in 11 randomized trials that compared BCS and mastectomy in terms of patient survival rate and quality of life in breast cancer and a second one including 909,077 patients in 35 trials both support that BCS post-NACT does not negatively impact long-term survival in breast cancer patients, even in cT3-4 breast cancer, supporting the use of BCS as a safe option. 

Some limitations must be declared, mainly the condition of a retrospective study based on chemotherapeutic regimens that have been surpassed nowadays and with a relatively small number of cases when divided by molecular surrogate subtypes. However, this is the result of a series with a long follow-up time that may be outdated in the chemotherapeutic schemas.

## 4. Conclusions

In conclusion, response to NACT plays a crucial role in assessing long-term patient prognosis. The important message to patients and clinicians for well-informed decision making is that an initial poor prognosis such as a triple-negative molecular subtype can be improved to an excellent prognosis if pCR is achieved, as has been demonstrated in the present data. 

Despite the mentioned limitations, we hope our data can help clinicians to better discuss with their patients the benefits of primary chemotherapy, placing a special emphasis on the prognostic value of response to neoadjuvant chemotherapy. 

## Figures and Tables

**Figure 1 cancers-16-02421-f001:**
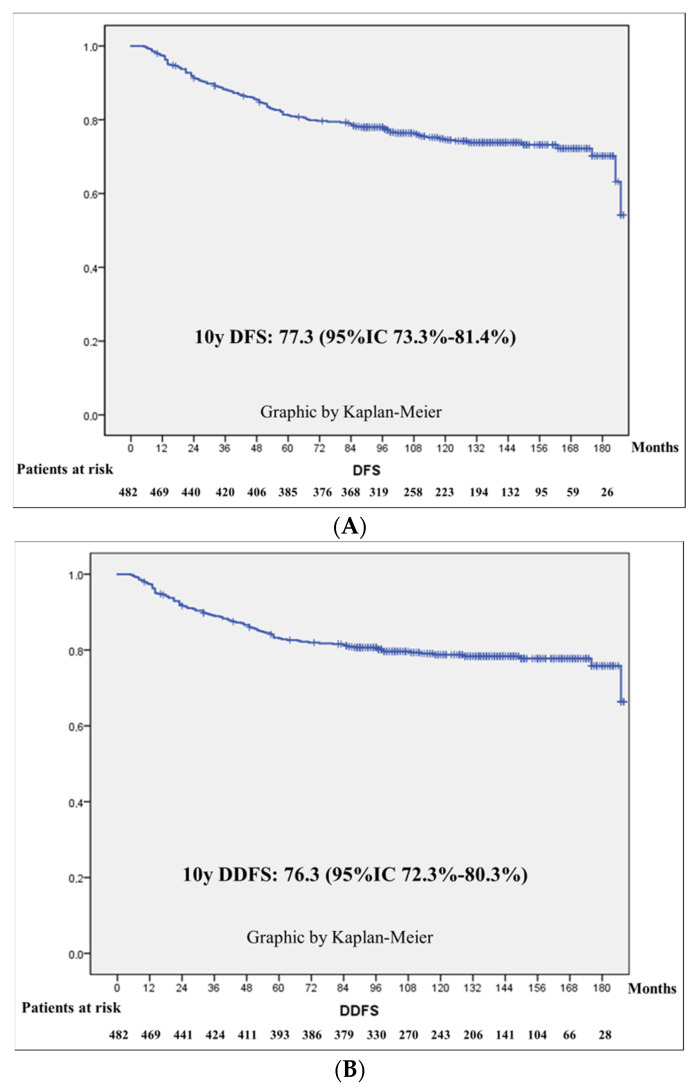
DFS, DDFS, BCSS and OS. Kaplan–Meier curves: (**A**) Disease-free survival (DFS). (**B**) Distant disease-free survival (DDFS). (**C**) Breast-cancer-specific survival (BCSS). (**D**) Overall survival (OS).

**Figure 2 cancers-16-02421-f002:**
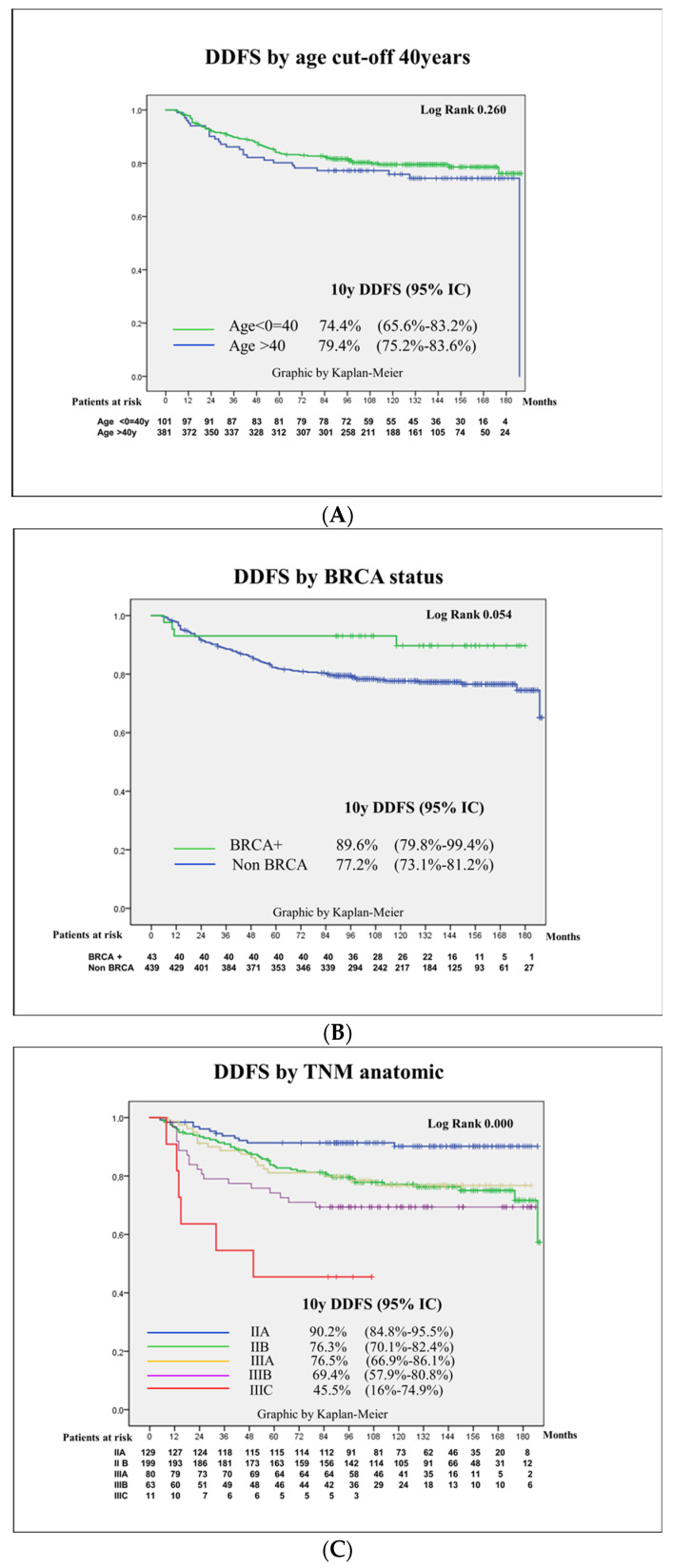
Kaplan–Meier curves for distant disease-free survival according to basal patient and tumor characteristics: age cut-off of 40, BRCA status, anatomic TNM, pathology subtype, molecular surrogate subtype and TILs at a 20% cut-off. Kaplan–Meier curves for distant disease-free survival (DDFS): (**A**) Age cut-off 40. (**B**) BRCA carriers. (**C**) Anatomic TNM. (**D**) Pathology subtype. (**E**) Molecular surrogate subtype. (**F**) TIL cut-off of 20%.

**Figure 3 cancers-16-02421-f003:**
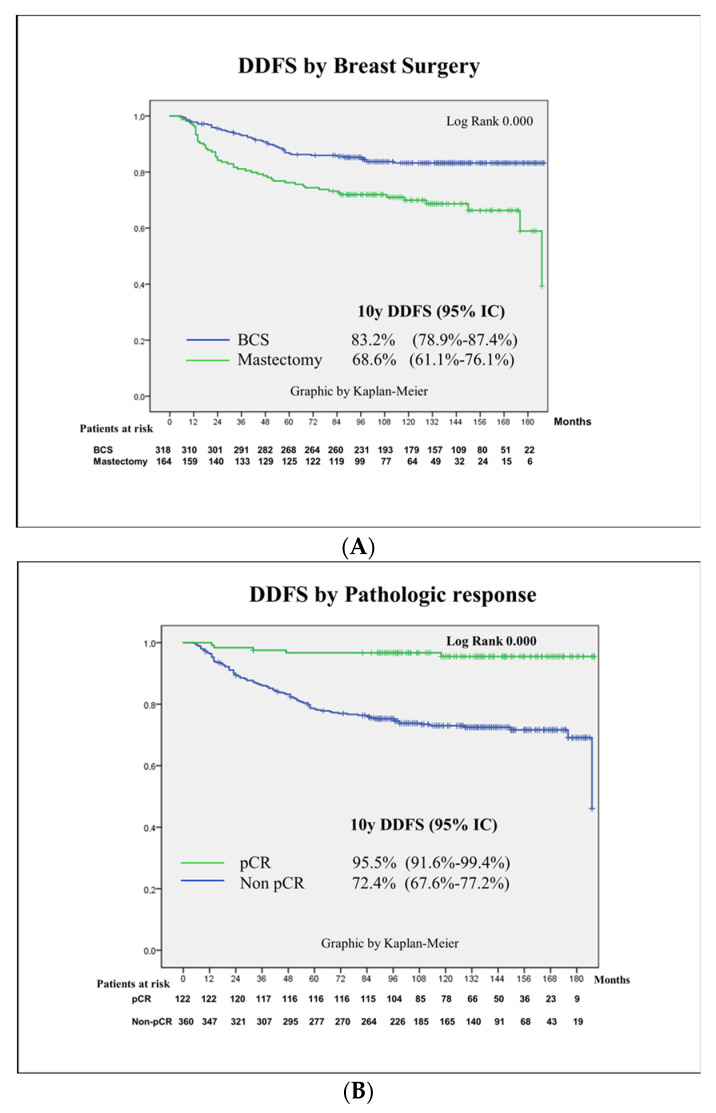
Kaplan–Meier curves according to surgical approach and pathological findings. Kaplan–Meier curves for distant disease-free survival (DDFS): (**A**) Breast surgery. (**B**) Pathologic complete response (pCR). (**C**) Residual cancer burden (RCB). (**D**) Vascular invasion.

**Figure 4 cancers-16-02421-f004:**
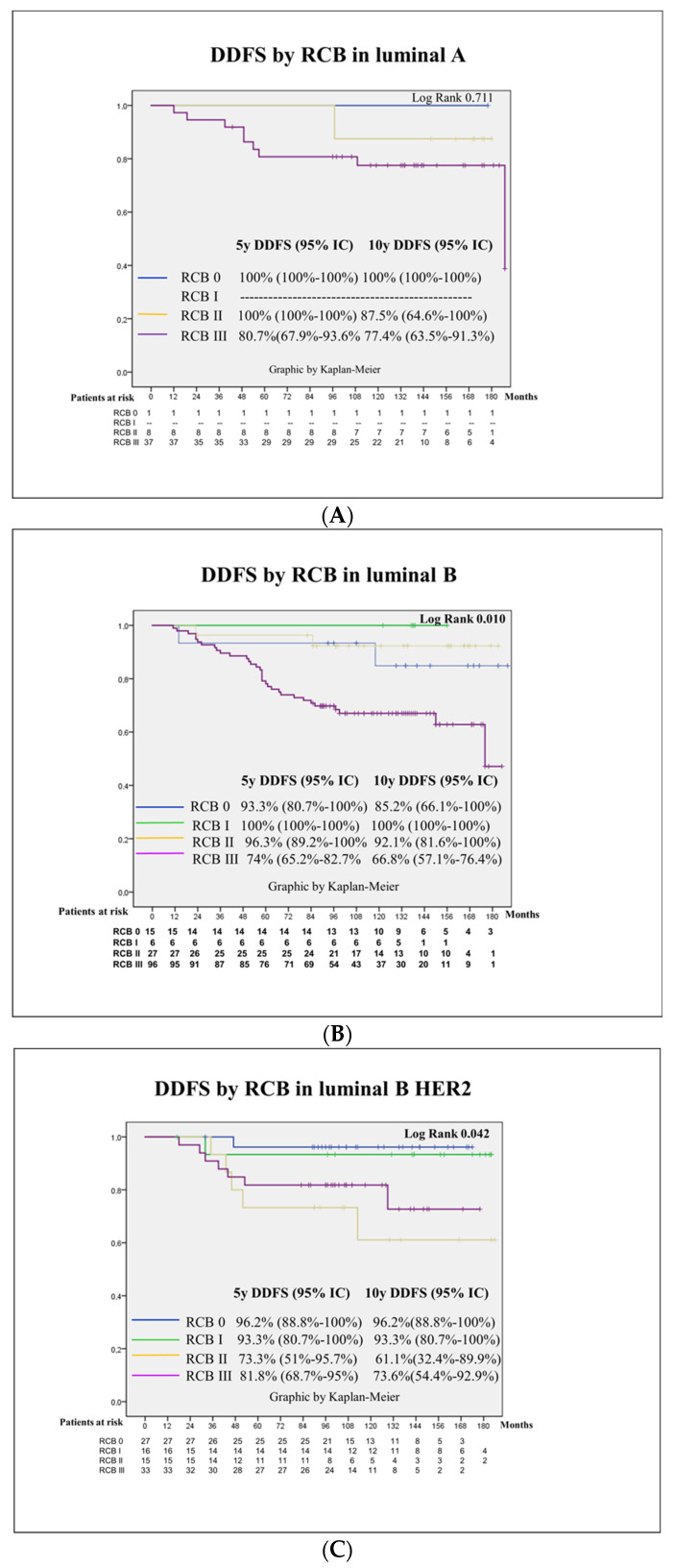
Kaplan–Meier curves according to RCB in each molecular surrogate subtype. Kaplan–Meier curves for distant disease-free survival (DDFS): (**A**) Luminal A-like. (**B**) Luminal B-like. (**C**) Luminal B HER2. (**D**) HER2. (**E**) Triple-negative breast cancer (TNBC).

**Table 1 cancers-16-02421-t001:** Patient and tumor characteristics.

N: 482	N (%)
Age 50 Years (SD 12.6)	
Age (years)	
≤40	101 (21%)
>40	381 (79%)
Menopausal status	
Pre	247 (51.2%)
Post	235 (48.8%)
BRCA carriers	
Yes	43 (8.9%)
No	439 (91.1%)
TNM anatomic	
IIA	129 (26.8)
IIB	199 (41.3)
IIIA	80 (16.6)
IIIB	63 (13.1)
IIIC	11 (2.3)
TNM prognostic	
IB	50 (10.4)
IIA	140 (29)
IIB	111 (23)
IIIA	48 (10)
IIIB	102 (21.2)
IIIC	31 (6.4)
Pathology subtype	
Ductal	459 (95.2)
Lobular	14 (2.9)
Others	9 (1.9)
Grade	
I	20 (4.1)
II	194 (40.2)
III	254 (52.7)
Ki 67	
≤30	203 (42.1)
>30	279 (57.9)
Molecular surrogate subtype	
Luminal A-like	46 (9.5)
Luminal B-like	144 (29.9)
LuminalBHER2	91 (18.9)
HER2	78 (16.2)
TNBC	123 (25.5)
TILs	
≤20	295 (61.2)
>20	187 (38.8)

Footnote: TILs: stromal tumor-infiltrating lymphocytes.

**Table 2 cancers-16-02421-t002:** Pathological, surgical and survival outcomes.

	N (%)
pCR	
Yes	122 (25.3)
No	360 (74.7)
RCB	
0	122 (25.3)
I	57 (11.8)
II	86 (17.8)
III	217 (45)
Vascular invasion	
No	376 (78)
Yes	101 (21)
Missing	5 (1)
Breast surgery	
Conservative	318 (66)
Mastectomy	164 (34)
Recurrences	
No	356 (73.9)
Contralateral	6 (1.2)
Local	54 (11.2)
Systemic	103 (21.4)
Deaths	
Breast cancer	78 (16.2)
Other cancers	9 (1.9)
Other causes	21 (4.3)
Total	108 (22.4)

Footnotes: pCR: pathologic complete response. RCB: residual cancer burden.

**Table 3 cancers-16-02421-t003:** Prognostic factors for patient survival. Univariate and multivariate analyses.

	Events		HR Univariate	HR Multivariate	*p*
Age (years)		0.223	*p*		
≤40	26 (25.7)				
>40	77 (20.2)				
Menopausal status		1			
Pre	53 (21.5)		0.9 (0.6–1.4)		
Post	50 (21.3)		Ref		
BRCA carriers		0.050			
Yes	4 (9.3)		0.3 (0.1–1.05)		
No	99 (22.6)		Ref		
TNM anatomic		0.000			
IIA	12 (9.3)		Ref		
IIB	48 (24.1)		2.6 (1.4–5.0)		
IIIA	18 (22.5)		2.5 (1.2–5.3)		
IIIB	19 (30.2)		3.8 (1.8–7.9)		
IIIC	6 (54.5)		9.6 (3.6–25.9)		
TNM prognostic		0.000			
IB	7 (14)		Ref		
IIA	25 (17.9)		1.2 (0.5–2.9)		
IIB	11 (9.9)		0.6 (0.2–1.7)		
IIIA	17 (35.4)		2.8 (1.1–6.8)		
IIIB	31 (30.4)		2.4 (1.0–5.6)		
IIIC	12 (38.7)		3.8 (1.5–9.8)		
Pathology subtype		0.003			
Ductal	92 (20)		Ref	Ref	
Lobular	8 (57.1)		3.6 (1.7–7.5)	4.4 (1.9–10.1)	**0.000**
Others	3 (33.3)		1.8 (0.5–5.6)	1.1 (0.3–3.8)	
Grade		0.175			
I	3 (15)		Ref		
II	49 (25.3)		1.7 (0.5–5.6)		
III	47 (18.3)		1.3 (0.4–4.1)		
Ki 67		0.125			
≤30%	49 (24.1)		Ref		
>30%	54 (19.4)		0.8 (0.5–1.2)		
Molecular surrogate subtype		0.074			
Luminal A-like	10 (21.7)		Ref	Ref	
Luminal B-like	37 (25.7)		1.2 (0.6–2.5)	1.4 (0.6–3.2)	
LuminalBHEr2	14 (15.4)		0.7 (0.3–1.6)	2.1 (0.8–5.6)	
HER2	10 (12.8)		0.6 (0.2–1.4)	2.5 (0.9–7.3)	
TNBC	32 (26)		1.4 (0.7–2.8)	4.0 (1.3–11.8)	**0.012**
TILs		0.023			
≤20%	73 (24.7)		1.6 (1–2.4)		
>20%	30 (16)		Ref		
Breast surgery		0.000			
Conservative	51 (16)		Ref	Ref	
Mastectomy	52 (31.7)		2.1 (1.4–3.2)	2.2 (1.4–3.4)	**0.000**
pCR		0.000		^a^	
Yes	5 (4.1)		Ref		
No	98 (27.2)		7.8 (3.1–19.3)		
RCB		0.000			
0	5 (4.1)		Ref	Ref	
I	6 (10.5)		2.8 (0.8–9.2)	2.2 (0.6–7.3)	
II	17 (19.8)		5.4 (2.0–14.8)	4.4 (1.5–12.8)	**0.006**
III	75 (34.6)		10.3 (4.1–25.6)	8.0 (2.9–21.7)	**0.000**
Vascular invasion		0.000			
No	61 (16.2)		Ref	Ref	
Yes	41 (40.6)		3.1 (2.0–4.6)	2.4 (1.5–3.7)	**0.000**

^a^: grade of freedom limited because covariables are lineally dependent (RCB). Footnotes: HR: Hazard ratio; pCR: pathological complete response; RCB: residual cancer burden; TILs: stromal tumor-infiltrating lymphocytes.

## Data Availability

The data can be shared up on request.

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
