# Peer review of "Breast Cancer Patient’s Outcomes after Neoadjuvant Chemotherapy and Surgery at 5 and 10 Years for Stage II–III Disease"

_cancers, 2024, doi:10.3390/cancers16132421_

Round 1

Reviewer 1 Report

Comments and Suggestions for Authors

The study entitled " Patient outcomes at 5 and 10 years after neoadjuvant chemotherapy for stage II and III breast cancer " by Falo et al. evaluates the prognostic value of response to NACT in Stage II-III breast cancer patients.

Conducted as a retrospective study at Bellvitge University Hospital, the research included 482 breast cancer patients.

The study showed a 25.3% rate of pCR, varying significantly across molecular subtypes, with the highest pCR rates in HER2-positive and TNBC.

Long-term survival outcomes, including DFS and DDFS, were significantly influenced by achieving pCR, with notable differences across subtypes.

Multivariate analysis identified ductal subtype, molecular subtype (HER2 and TNBC), response to NACT (RCB score), and vascular invasion as key prognostic factors.

The study has some strong points: first, the very long follow-up period; secondly, the methods are very clearly described in details.

However, I have some comments which will improve the manuscript:

- I suggest to modify the title in "Breast cancer patients outcomes after neoadjuvant chemotherapy and surgery at 5 and 10 years for Stage II-III disease";

- The first sentence of the Introduction is very true; however, my suggestion is to incorporate these studies PMID: 38539504, PMID: 30348601 to include recent literature and improve your manuscript quality;

- It is certainly true that NACT has been used as a prognostic tool to guide prognosis/adjuvant treatment of breast cancer patients and achieving a pCR is certainly a favourable prognostic sign and vascular invasion is a sign of poor prgnosis. Similarly, these studies PMID: 37001289, PMID: 30006427 reached similar results. Please cite them to give more context to your research and improve your manuscript;

- Line 108 " when a mastectomy was mandatory..." Please explain why and when a mastectomy was mandatory;

- "Line 123, "worst-case scenario". Please remove this expression. It is not very academic;

- Tables need revision of style. Authors can also incorporate/merge them;

- Figures are too small for readers;

- Please move limitations BEFORE conclusions.

Author Response

  1. Answer to reviewer 1:
    1. Reviewer sentence: I suggest to modify the title in "Breast cancer patients outcomes after neoadjuvant chemotherapy and surgery at 5 and 10 years for Stage II-III disease". Response: Thank you very much for your suggestions, we agree with your comments, therefore we have changed the tittle in line 2 and 3:  “Patient outcomes at 5 and 10 years after neoadjuvant chemotherapy for stage II and III breast cancer” by your proposal: Breast cancer patients outcomes after neoadjuvant chemotherapy and surgery at 5 and 10 years for Stage II-III disease.
    2. Reviewer sentence: The first sentence of the Introduction is very true; however, my suggestion is to incorporate these studies PMID: 38539504, PMID: 30348601 to include recent literature and improve your manuscript quality. Response: We appreciate your comment very much, therefore we have incorporated these two valuable studies and some of the references of those studies in the first paragraph of the introduction line 66 to 81: Notably, this approach has greatly facilitated the shift from mastectomy to breast conservative surgery (BCS) as a surgical option, particularly in patients with initially diagnosed cT3-4 breast cancer. A recently publication by Tinterri et al (4) specifically looked at patients with cT3-4 breast cancer. The aims of that study was to identify predictors of breast conservation in cT3-4 breast cancer and compared the long-term oncological outcomes between BCS and mastectomy. The authors identified the absence of vascular invasion, smaller tumor size post NACT and achieving a complete pathological response of the primary tumor as the key predictors for breast conservation. In addition, after a follow up of 70 months (range,52-185), they demonstrated that BCS post-NACT does not negatively impact long-term oncological outcomes, supporting its use as a safe option for patients with cT3-4 breast cancer. Data from the Dutch Breast Audit published by Spronk et al (5) confirms the safety of BCS after NACT. Those authors analyzed trends in the use of NACT and the impact on surgical outcomes. In this audit between 2011 and 2016, the use of NACT increased from 9% to 18% and BCS after NACT increased from 43% to 57%. Prognostic factors associated with invasive margin rate were lobular invasive breast cancer and a hormone receptor positive status. When compared with results of upfront BCS, the authors confirmed that BCS after NACT compared to primary BCS leads to equal surgical outcomes for cT2 and improved surgical outcomes for cT3 breast cancer patients. Many other studies have consistently demonstrated that BCS does not compromise recurrence and survival rates in patients with breast cancer treated with NACT(6–9).
    3. Reviewer sentence It is certainly true that NACT has been used as a prognostic tool to guide prognosis/adjuvant treatment of breast cancer patients and achieving a pCR is certainly a favorable prognostic sign and vascular invasion is a sign of poor prognosis. Similarly, these studies PMID: 37001289, PMID: 30006427 reached similar results. Please cite them to give more context to your research and improve your manuscript. Response Completely agree, accordingly we have enlarged the introduction with a more detailed description on pCR and non pCR grading and relation to patient outcome: in the third paragraph of the introduction lines 97 to 118: NACT provides the unique opportunity to assess response to treatment after months rather than years of follow-up. Achieving a pathologic complete response (pCR) has been associated with significant improvement in disease free survival (DFS) and overall survival (OS), and has therefore become as a surrogate end-point for long-term survival and primary aim in numerous clinical trials(13)(14)(15)(16). In a meta-analysis of Broglio et al(17) including 36 randomized clinical trials (RTCs) including stage I-III HER2 positive breast cancer patients treated with NACT, showed a substantial improvement in event free survival for pCR vs non pCR with a HR, 0.37 (95% PI 0.32-0.43), this association was greater for patients with hormone receptor negative disease with a HR, 0.29 (95% PI 0.24-0.36) than hormone receptor-positive disease with a HR, 0.52 (95%PI 0.40-0.66). However, in a systematic review and meta-analysis, Conforti et al(18) found a weak association between the relative risk for pCR and hazard ratio (HR) for both disease-free survival and overall survival. In addition, most patients do not experience a complete pathological response to primary chemotherapy and the significance of lesser degrees of histological response is uncertain and the prognostic significance is unknown. Several efforts have been made to evaluate new histological grading systems, such as the Miller/Payne grading system from the University of Aberdeen(19) or the neoadjuvant response index from the Nederland(20). Gentile et al from Milan(21), reported the pathological response and residual tumor cellularity after neoadjuvant chemotherapy in relation to patient prognosis. These authors found statistically significant longer DFS, DDFS and OS in patients with pCR and with a residual tumor cellularity inferior to 40%. In 2015 the Breast International Group-North American Breast Cancer Group (the BIG-NABCG) published the recommendations for standardized pathological characterization of residual disease for neoadjuvant clinical trials(22). Recommendations included multidisciplinary communication, clinical marking of the tumor site with clips, and radiologic, photographic or pictorial imaging of the sliced specimen to map the tissue sections and reconcile macroscopic and microscopic findings. The information required to define pCR (including carcinoma in situ or not), residual disease stage using the current AJCC/UICC system, and the Residual Cancer Burden System were recommended for quantification of residual disease in clinical trials). We have also included a sentence in the discussion following your recommendation in line 576 to 580: According to the results of the literature(13)(14)(15)(16)(17), in our series, those patients who achieved a pCR showed better survival outcomes related to those patients that did not. The estimated 10y DDFS for patients achieving pCR was 95.5% (95% CI 91.6%-99.4%) and for non-pCR was 72.4% (95% CI 67.6%-77.2%) similar to results from the series of Gentile et al(21). And in line 652-661: Breast conserving surgery was achieved in 66% of this series that included 32% of locally advanced breast cancer. Survival was not affected by this decision, in fact, performing a mastectomy doubled the risk of distant recurrence in comparison to BCS, p=0.000. Results that confirm the already mentioned studies from Tinterri et al(4), the Dutch Breast Audit(5), and many others(6–9) including two recent meta-analysis(49,50). One including 34.169 patients in 11 randomized trials that compared BCS and mastectomy in terms of patient survival rate and quality of life in breast cancer and the second one, including 909077 patients in 35 trials, all support that BCS post-NACT does not negatively impact long-term survival in breast cancer patients, even in cT3-4 breast cancer, supporting the use of BCS as a safe option. 

  1. Reviewer sentence Line 108 " when a mastectomy was mandatory..." Please explain why and when a mastectomy was mandatory. Response: we agree and modified in line 156-158: When a mastectomy was mandatory, in multicentric tumors, inflammatory cases (T4d) or if the tumor volume and breast size precluded breast conservation with satisfactory aesthetic outcomes or offered to BRCA carriers, immediate reconstructive options were offered to the patient, predominantly with autologous flaps to avoid the development of capsular fibrosis with silicone implants
  2. Reviewer sentence "Line 123, "worst-case scenario". Please remove this expression. It is not very academic;. Response We agree with this comment and this expression has been removed from line 175
  3. Reviewer sentence Tables need revision of style. Authors can also incorporate/merge them; Response: Tables are re-buit to improve style, thank you for pointing this out
  4. Reviewer sentence Figures are too small for readers; Response Figures will be enlarged and separated to improve visualization for readers
  5. Reviewer sentence Please move limitations BEFORE conclusions. Response We agree with this comment, and we have accordingly modified the order of the limitations before the conclusions that now are in lines 663 to 667
  6. I extended the acknowledgments to the cancer reviewers. Thank you very much for your recommendations for improving the manuscript. In line 678 and 679.

Reviewer 2 Report

Comments and Suggestions for Authors

1. The drawings are of very poor quality, it is impossible to see the inscriptions, they need to be redone.

2. There are no detailed captions; several figures (A, B, C, etc.) are not deciphered.

3. In table 3 in the last line, typo 31 - is it 3.1?

4. Did any patients receive adjuvant chemotherapy and/or radiation therapy? Is this factor taken into account in assessing survival rates? or was there only NACT and surgery? Does the extent of surgical intervention affect the prognosis?

5. Is there a relationship between NACT drugs and survival rates? Treatment regimen, number of courses, etc.?

6. I did not see in the analysis the age of the patients, menopausal status, or concomitant pathologies that could affect the outcome of treatment.

Author Response

  1. Answer to reviewer 2:
    1. Reviewer sentence: The drawings are of very poor quality, it is impossible to see the inscriptions, they need to be redone. Response: I completely agree, and I have redone them.
    2. Reviewer sentence: There are no detailed captions; several figures (A, B, C, etc.) are not deciphered. Response: Thank you for pointing this out, I have modify them all.
    3. Reviewer In table 3 in the last line, typo 31 - is it 3.1? Response : thank you very much for revising so in detail all the numbers, effectively it was an error. I amended accordingly: 1 (2.0-4.6)
    4. Reviewer sentence. Did any patients receive adjuvant chemotherapy and/or radiation therapy? Is this factor taken into account in assessing survival rates? or was there only NACT and surgery? Does the extent of surgical intervention affect the prognosis? Response: Most of the patients did not received adjuvant chemotherapy except capecitabine in those TNBC without pCR only since 2015 explained in line 173-174: TNBC cases with residual disease after NACT, were offered adjuvant capecitabine since 2015. When we analyzed survival according to radiotherapy, the results were linked to the stage of the disease not in relation to radiotherapy administration and for this reason I have not mentioned this result. In reference to the extent of surgical intervention, I appreciate your request very much and I have added a Kaplan Meier curve according to the extent of surgical intervention. Figure 3 A. This data was very interesting and it is added at results line 542 and to the discussion lines 652 to 661.
    5. Reviewer sentence Is there a relationship between NACT drugs and survival rates? Treatment regimen, number of courses, etc.? Response I appreciate very much this comment, we did a great effort to calculate de dose dense of each drug for relating with prognosis, but the results were incomplete, for this reason we decided to review again case per case, drug by drug and we could probably write a manuscript focused in this interesting point in a near future.
    6. Reviewer sentence I did not see in the analysis the age of the patients, menopausal status, or concomitant pathologies that could affect the outcome of treatment. Response: thank you for pointing this out, I have included those analyses according to age cut-off 40 and menopausal status and I have incorporated them in table 3 and have selected age cut-off 40 for Fig 2. Concomitant pathologies were not always adequately recorded in the clinical records, and I prefer not to do the analysis in this poor data. It will be worthy to explore clinical reviews again patient by patient focusing on that point as well as in late toxicity caused by treatment such as cardiotoxicity or acute leukemia and write a separate manuscript.
    7. I extended the acknowledgments to the cancer reviewers. Thank you very much for your recommendations for improving the manuscript. In line 678 and 679

Reviewer 3 Report

Comments and Suggestions for Authors

The manuscript described a retrospective observational study examining 482 stage II and III breast cancer patients treated with neoadjuvant chemotherapy (NACT) from 2008 to 2016. The authors examined the outcomes, including disease-free survival (DFS), distant disease-free survival (DDFS), breast cancer-specific survival (BCSS), overall survival (OS), pathologic complete response (pCR), residual cancer burden (RCB), vascular invasion, locoregional and distant recurrences, and overall mortality. They discovered that HER2-positive and TNBC subtypes showed distinct responses and survival outcomes based on NACT response. Additionally, luminal subtypes demonstrated lower pCR rates and late recurrence patterns compared to HER2-positive and TNBC. Overall, the study underscored that advances in NACT regimens have significantly improved survival outcomes, particularly for HER2-positive and TNBC subtypes. Furthermore, newer therapies like immune checkpoint inhibitors and CDK 4/6 inhibitors should be incorporated for luminal subtypes. In summary, the manuscript provides a detailed analysis of NACT outcomes in stage II and III breast cancer, highlighting significant advancements in treatment strategies and their impact on long-term survival outcomes across different molecular subtypes. Here are potential areas for improvement:

1, Provide more detailed information on patient selection criteria, especially how breast cancer subtypes were identified.

2. Improve clarity and resolution of figures.

3, Any available genomic profiling information from these patients? If yes, their potential implications for treatment response and prognosis could be discussed.

4, Perform additional subgroup analysis to evaluate treatment efficacy in these distinct patient groups. For example, conduct analyses based on age, or menopausal status.

5, Any analyses have been conducted to examine changes in biomarker expression (e.g., Ki-67, HER2 status) before and after neoadjuvant chemotherapy and their impact on treatment response and survival outcomes?

Comments on the Quality of English Language

Minor improvements in grammar, syntax, and sentence structure could enhance the manuscript.

Author Response

  1. Answer to reviewer 3:
    1. Reviewer sentence: Provide more detailed information on patient selection criteria, especially how breast cancer subtypes were identified. Response: Thank you for pointing this out, I accordingly have included a sentence in the inclusion criteria of patients describing the classification of patients in the different molecular surrogated subtypes according to st Gallen 2013 in line 135 to 143: Patients were classified into five molecular surrogated subtypes according to St Gallen 2013 in luminal A-like (estrogen receptor (ER) positive, progesterone receptor (PR) positive, HER2 negative and ki 67 <20%); luminal B-like (RE positive, PR positive or negative, HER2 negative and ki 67 over 20%); luminal B Her2 ( RE positive, PR positive or negative, HER2 positive); HER2 (RE negative, PR negative, HER2 positive) and triple negative (TNBC) (RE negative, PR negative and HER2). ER and PR were considered positive if expressed in 10% or more of the tumor cells. HER2 positive tumor were considered those with immunohistochemistry 3+ or 2+ with gene amplification by fluorescent in situ hybridization (FISH).
    2. Reviewer sentence: Improve clarity and resolution of figures. Response: Thank you for pointing this out, I have modified and sent them all.
    3. Reviewer Any available genomic profiling information from these patients? If yes, their potential implications for treatment response and prognosis could be discussed .Response : thank you very much for this comment. Accordingly, I have reviewed all the clinical records in search of such information, and I have added an analysis to check their potential implications for treatment response and prognosis. The information is added in table 1. In theu univariate and multivariate analysis described in table 3, and I have run a Kaplan Meier according to the BRCA status Fig 2B. In the text the information about BRCA carriers have been added in lines 192-193; 223-224; 239-240; 249-250; 315; 552-555 and in the discussion lines 640-642.
    4. Reviewer sentence: Perform additional subgroup analysis to evaluate treatment efficacy in these distinct patient groups. For example, conduct analyses based on age, or menopausal status. Response: thank you for pointing this out, I have included those analyses that are done according to age and menopausal status but without statistical difference, and I have incorporated them in table 1 and 3, following your recommendation. In addition a Kaplan Meier curves have been done according to age cut-off 40 (Fig 2A)
    5. Reviewer sentence: Any analyses have been conducted to examine changes in biomarker expression (e.g., Ki-67, HER2 status) before and after neoadjuvant chemotherapy and their impact on treatment response and survival outcomes? Response: thank you for pointing this out. Some years ago, we examined changes in biomarker expression of this series (estrogen receptor, HER2) but not significant changes were seen, and the few cases with a therapeutic impact were those that shift from negative to positive. According to these results, in the pathology department only those factors that were negative at baseline are retested in the surgical specimen, to find those patients with heterogeneous tumor that could benefit from a positive result. In relationship with changes in ki 67, our pathologists only report it after hormonal treatment, and for this reason, I am sorry but I cannot offer this data.
    6. I extended the acknowledgments to the cancer reviewers. Thank you very much for your recommendations for improving the manuscript. In line 678 and 679

Round 2

Reviewer 1 Report

Comments and Suggestions for Authors

The manuscript can be accepted in the present form

Reviewer 2 Report

Comments and Suggestions for Authors

I have no further comments on the manuscript.